# *Arbutus unedo* L. Fractions Exhibit Chemotherapeutic Properties for the Treatment of Gastrointestinal Stromal Tumors

**DOI:** 10.3390/plants13091201

**Published:** 2024-04-25

**Authors:** Aldo Di Vito, Manuela Mandrone, Ilaria Chiocchio, Francesca Gorini, Gloria Ravegnini, Emma Coschina, Eva Benuzzi, Simona Trincia, Augusto Hubaide Nozella, Trond Aasen, Cinzia Sanna, Fabiana Morroni, Patrizia Hrelia, Ferruccio Poli, Sabrina Angelini

**Affiliations:** 1Department of Pharmacy and Biotechnology, University of Bologna, 40126 Bologna, Italy; aldo.divito2@unibo.it (A.D.V.); manuela.mandrone2@unibo.it (M.M.); ilaria.chiocchio2@unibo.it (I.C.); francesca.gorini3@unibo.it (F.G.); gloria.ravegnini2@unibo.it (G.R.); emma.coschina2@unibo.it (E.C.); eva.benuzzi2@unibo.it (E.B.); simona.trincia2@unibo.it (S.T.); augusto.nozella2@unibo.it (A.H.N.); fabiana.morroni@unibo.it (F.M.); ferruccio.poli@unibo.it (F.P.); s.angelini@unibo.it (S.A.); 2Patologia Molecular Translacional, Vall d’Hebron Institut de Recerca (VHIR), CIBERONC, 08035 Barcelona, Spain; trond.aasen@vhir.org; 3Department of Life and Environmental Sciences, University of Cagliari, 091243 Cagliari, Italy; cinziasanna@unica.it; 4Inter-Departmental Center for Health Sciences & Technologies, CIRI-SDV, University of Bologna, 40126 Bologna, Italy

**Keywords:** *Arbutus unedo* L., plant extract, bio-guided fractionation, gastrointestinal stromal tumor, GIST, chemotherapeutic properties

## Abstract

Novel treatments in gastrointestinal stromal tumors (GISTs) are essential due to imatinib resistance and the modest results obtained with multi-target tyrosine kinase inhibitors. We investigated the possibility that the hydroalcoholic extract from the leaves of *Arbutus unedo* L. (AUN) could harbor novel chemotherapeutics. The bio-guided fractionation of AUN led to a subfraction, FR2-A, that affected the viability of both imatinib-sensitive and -resistant GIST cells. Cells treated with FR2-A were positive for Annexin V staining, a marker of apoptosis. A rapid PARP-1 downregulation was observed, although without the traditional caspase-dependent cleavage. The fractionation of FR2-A produced nine further active subfractions (FRs), indicating that different molecules contributed to the effect promoted by FR2-A. NMR analysis revealed that pyrogallol-bearing compounds, such as gallic acid, gallic acid hexoside, gallocatechin, myricetin hexoside, and trigalloyl-glucose, are the main components of active FRs. Notably, FRs similarly impaired the viability of GIST cells and peripheral blood mononuclear cells (PBMCs), suggesting a non-specific mechanism of action. Nevertheless, despite the lack of specificity, the established FRs showed promising chemotherapeutic properties to broadly affect the viability of GIST cells, including those that are imatinib-resistant, encouraging further studies to investigate whether pyrogallol-bearing compounds could represent an alternative avenue in GISTs.

## 1. Introduction

Plants have been an inestimable source of anticancer compounds, providing approximately 50% of the approved chemotherapeutics [1,2,3]. In fact, natural compounds (NPs) show promising features, such as scaffold diversity and structural complexity, that make them ideal for drug discovery [4,5]. *Arbutus unedo* L. (*A. unedo*) is a Mediterranean plant used in traditional medicine to treat various illnesses, such as gastrointestinal, dermatological, cardio-vascular, and urological disorders, kidney diseases, or also used as a diuretic and antidiabetic, suggesting the presence of numerous NPs with promising pharmacological activities [6]. Regarding potential uses in cancer, studies have reported that NPs extracted from the entire plant, fruits, honey, and leaves could promote cytotoxicity in cellular tumor models. In particular, it has been recently reported that the leaf extract of *A. unedo* reduced the viability of U2OS, a cellular model of osteosarcoma, without toxicity in human umbilical vein endothelial (HUVEC) cells, a healthy cellular model [7]. However, further investigations are required to identify the NPs responsible for the observed anticancer activity.

GISTs are rare mesenchymal neoplasms caused by a gain of function mutation in KIT or PDGFRα, two tyrosine kinase receptors (TKRs). For this reason, they are treated with imatinib, a tyrosine kinase inhibitor (TKI) [8]. Although imatinib has significantly improved patient prognosis, providing remarkable amelioration of their life expectations, the therapy is not conclusive. Indeed, imatinib mainly promotes partial response or stable disease in most patients, while complete response is only observed in 5% of patients [9]. Moreover, imatinib treatment is frequently associated with drug resistance, mostly due to acquired secondary mutations in TKRs (50–60% of patients) [10,11], fostering the development and approvals of novel multi-target TKIs as further treatment lines [12]. Unfortunately, resistance has not been effectively treated with a strategy exclusively based on TKIs, which have provided a modest increase in progressive free survival (PFS). Preclinical studies have reported that imatinib resistance could be more multifaceted than initially hypothesized, supporting the research of novel compounds that could broadly target imatinib-resistant GIST cells [13]. Traditional chemotherapy could potentially represent an alternative strategy for treating GISTs. Indeed, the lack of cell specificity, a limitation of traditional chemotherapy, could be an advantage, potentially targeting both TKI-sensitive and -resistant subclones through a non-specific mechanism of action. However, the available chemotherapeutics, including those effective in other soft tissue sarcomas, such as doxorubicin, had already been tested in GISTs before imatinib approval, showing no effect [14,15]. Thus, no standard chemotherapy has been approved in GIST so far [16].

Therefore, considering that identifying novel chemotherapeutics could be promising for treating GISTs and the interesting anticancer properties of *A. unedo* already reported in the literature, we hypothesized that *A. unedo* could harbor NPs with chemotherapeutic properties. Therefore, we observed that an extract obtained from *A. unedo* leaves affected the viability of GIST cell lines characterized by different mutations and degrees of imatinib response. Furthermore, a bio-guided assay fractionation procedure was employed to investigate the different compounds’ contribution to the crude extract’s activity.

## 2. Results

### 2.1. The Crude Extract from the Leaves of A. unedo Impairs GIST Cell Viability and Triggers Apoptosis

The pharmacological effect of the hydroalcoholic crude extract from *A. unedo* leaves (hereafter referred to as AUN only) was first assessed in the KIT-mutated imatinib-sensitive cell lines GIST-882 and GIST-T1. In detail, cells were treated with three serial 1:2 dilutions of AUN starting from 200 µg/mL for 72 h. AUN treatment significantly decreased the viability in GIST-882 (Figure 1a) and GIST-T1 (Figure 1b) with respect to the control. In GIST-882, a significant 75% and ~25% reduction was observed at 200 µg/mL and 100 µg/mL concentrations, while no effect was observed at 50 µg/mL. Similarly, cell viability was significantly impaired by about 75% at 200 µg/mL in GIST-T1, while a non-significant reduction was observed after 100 and 50 µg/mL treatments.

Considering the doubling time of GIST-882 and GIST-T1, experimentally calculated as approximately 48–72 h, the effects observed following AUN treatment could be due to both the inhibition of cell proliferation and cell death. Interestingly, in the first gross investigation by flow cytometry, a further cell population with a lower cell size index and a slight increase in viability staining was observed in AUN-treated GIST-882 compared to the control (Appendix A). Additionally, this population was characterized by reduced nucleus staining, supporting the hypothesis that these cells contain degraded chromosomal DNA. The two features—small size and degraded DNA—suggested that AUN could stimulate cell death by triggering apoptosis. This hypothesis was confirmed using the Annexin-V/7-AAD assay in both cell lines. In particular, in GIST-882, 24 h of AUN treatment promoted a significant increase in early (38%, *p* < 0.0001) and late (41%, *p* < 0.0001) apoptotic cells compared to the control (early 21% and late 13%, respectively; Figure 2—upper panel). Analogously, although with faster kinetics, an increase in early and late apoptotic cells was also observed in GIST-T1 after 6 h of treatment (Figure 2—lower panel). In detail, we observed a significant increase in both early and late apoptosis compared to the control (31% vs. 15%; *p* < 0.0001 and 15% vs. 3%; *p* < 0.001, respectively).

### 2.2. Preliminary Steps of Bio-Guided Fractionation

AUN was first fractionated using a liquid/liquid partition to preliminary separate the compounds based on their polarity. Hence, three fractions were obtained from AUN: FR1, which contained phytochemicals extracted by chloroform (yield about 0.33% *w*/*w*); FR2, which collected those that are instead extracted by ethyl acetate (yield about 8.8% *w*/*w*); and FR3, retaining the more hydrophilic phytochemicals in water. Each fraction was then tested in GIST-882 and compared with AUN to identify the fraction where active phytochemicals were mainly distributed. As shown in Figure 3a, GIST-882 viability was impaired primarily by FR2, promoting a significant reduction of about 90% (*p* < 0.0001), which was even more intense than AUN. Approximately a 30% viability reduction was observed in the FR3-treated sample (*p* < 0.05), while FR1 promoted no effect. Overall, the data indicated that following liquid/liquid partition, active phytochemicals were mainly collected and enriched in FR2 with respect to FR1 and FR3. Furthermore, the percentage of early apoptotic cells observed through the Annexin-V/7-AAD assay corroborated that the active proapoptotic phytochemicals were principally distributed in FR2 (Figure 3b). Similar results were observed in GIST-T1 cells, and in particular, cell viability was significantly reduced by 75% (*p* < 0.001) after 24 h of FR2 treatment (Appendix A).

Altogether, these data strengthen the fact that the active phytochemicals were extracted into FR2, and they can similarly target imatinib-sensitive cellular models characterized by different primary KIT mutations. The above data led us to consider FR2 as the fraction harboring the phytochemicals of interest and the starting material for a second step of bio-guided fractionation. In particular, it was subjected to reverse-phase medium-pressure liquid chromatography (MPLC), resulting in four FR2 subfractions (FR2-A, FR2-B, FR2-C, and FR2-D; Appendix A) generated based on the chromatogram. Afterwards, the activity of the four subfractions was tested in GIST-882, with cell viability evaluated after 24 h of treatment with two different concentrations (12.5 µg/mL and 50 µg/mL) compared to the control (Figure 4). In total, 50 µg/mL of FR2-A significantly impaired cell viability (>95%; *p* < 0.0001), similarly to that observed after treatment with 200 µg/mL of FR2. Moreover, at the same concentration, FR2-B and FR2-C promoted a significant 40% reduction in cell viability (*p* < 0.001). At the lowest concentration (12.5 µg/mL), a significant reduction (*p* < 0.01) was observed after FR2-A treatment only; indeed, no effect of FR2 and the other FR2 subfractions was observed. Overall, this result led us to identify FR2-A as the most active among the FR2 subfractions. Consistent with this hypothesis, FR2-A also affected GIST-T1 viability, and the calculated IC50 was around 33 µg/mL in both cellular models (Appendix A). Consequently, FR2-A was further investigated regarding its effect on imatinib-resistant cells and its mechanism of action.

### 2.3. FR2-A Targets GIST Cells through a KIT-Independent Mechanism

To further investigate FR2-A activity, GIST-48 and GIST-48b, recognized as imatinib-resistant cells [17], were treated with the FR2-A subfraction. Notably, cell viability was impaired in both models, even at a higher IC50 compared to GIST-882 and GIST-T1 (Figure 5a,b). Interestingly, GIST-48b cells, unlike GIST-882, -T1, and -48 cells, are characterized by no detectable level of the oncoprotein KIT (Figure 5c), suggesting that the mechanism of action of FR2-A was KIT-independent and could provide an alternative treatment to TKIs.

To further deepen the FR2-A mechanism of action, GIST-882 cells were treated in a time-course experiment of 48 h, with 33 and 66 µg/mL FR2-A as well as with the gold standard imatinib (Figure 6a). In particular, 66 µg/mL of FR2-A reduces viability by 50% about 4 h after treatment, while 33 µg/mL of FR2-A requires about 6 h. In addition, FR2-A impaired almost entirely cell viability at 66 µg/mL, whereas the 33 µg/mL treatment did not reduce the viability by more than 50%. Thus, it could be hypothesized that FR2-A affects GIST-882 viability in a dose- and time-dependent manner. Regarding imatinib treatment, cell viability was significantly affected (*p* < 0.05), but only after 48 h, which may be due to reduced proliferation rather than reduced cell viability. In line with these observations, Annexin-V/7-AAD staining confirmed that imatinib promoted a slight increase (23.4% vs. 15.6%; *p* < 0.05) in early apoptotic cells compared to the control at 48 h (Figure 6b,c). On the contrary, both concentrations of FR2-A gave rise to a remarkably higher increase in early apoptotic cells compared to the control, as previously shown with AUN and FR2. A total of 66 µg/mL of FR2-A led to a gradual and significant increase in the early apoptotic population starting from 2 h after treatment (*p* < 0.01), leading to a percentage of approximately 50–60% 48 h after treatment (*p* < 0.001). The lower concentration, 33 µg/mL, required about 6 h to show a significant increase in early apoptotic cells, inducing the increase to about 40% at 48 h (Figure 6b,c).

Considering that GIST-882 viability was quickly impaired 2 h after treatment, we focused on the mechanism of action of FR2-A at early time points. Specifically, the activation of traditional apoptosis pathways based on the cleavage of caspase-3 or caspase-9 was monitored. Although FR2-A promoted the rise of an Annexin V-positive and 7-AAD-negative population, commonly defined as early marks of apoptosis, no caspase activation was observed. Failure to detect the cleavage of PARP-1, a well-recognized downstream target of caspase-3, confirmed the absence of traditional caspase-3 activation. Nevertheless, treatment with 66 µg/mL of FR2-A determined a time-dependent downregulation of PARP-1. In detail, the level of PARP-1 was significantly reduced by about 50% after 2 h (*p* < 0.01), and even by about 85% after 3 h (*p* < 0.001), supporting the hypothesis that, at least in the late stage, the pharmacological effect could be strictly related to PARP-1 downregulation (Figure 7).

### 2.4. Further Fractionation of FR2-A

Since FR2-A was associated with promising results, notably impairing the viability of GIST cells independently from KIT expression and mutational patterns, its chemical composition was analyzed by ^1^H NMR to identify the active compounds. Intense spectral signals reveal that the most represented compound in FR2-A was arbutin in its β anomeric form (Appendix A). Therefore, GIST-882 cells were treated with pure β-arbutin at 200 µg/mL. However, no significant effect was detected on GIST-882 cell viability and apoptosis up to 72 h after treatment, indicating that β-arbutin was not the compound responsible for FR2-A activity (Appendix A). In view of this result, FR2-A was further fractionated using size-exclusion chromatography (SEC), yielding 61 analyzable subfractions. Among them, nine maintained the pharmacological activity in GIST-882 and GIST-T1, suggesting that more than one phytochemical was responsible for the FR2-A effect. Specifically, 30 µg/mL of 2A-29, 2A-30, 2A-31, 2A-35, 2A-36, 2A-60, and 2A-61 significantly reduced the viability of both cell lines up to the limit of assay detection (*p* < 0.0001; Figure 8a); the same dose of 2A-48 and 2A-49 provided a significant viability reduction of no more than 60% (*p* < 0.0001). Interestingly, 2A-35 and 2A-36 subfractions were also active at 6 µg/mL, thus representing the most active ones (Figure 8b). To make reading easier, the figures only include the nine active subfractions delimitated by the first available adjacent weakly or non-active subfractions tested (i.e., with available material). Interestingly, groups of numerically adjacent subfractions had similar activity, suggesting that they could harbor the same bioactive compound(s). Based on this assumption, we combined active contiguous subfractions into four clusters (C.2A-29-31, C.2A-35-36, C.2A-48-49, and C.2A-60-61).

### 2.5. FR2-A Clusters Reduce Cell Viability in GISTs and Lymphocytes

The toxicity of FR2-A-derived clusters was evaluated in peripheral blood lymphocytes (PBMCs). In particular, PBMCs and GIST-T1 were treated with the established four clusters at 30 µg/mL, while the 2A-20 subfraction, previously evaluated as inactive in GIST cells, was used as a negative control (Figure 9a). All clusters significantly reduced the viability of both GIST-T1 and PBMCs. In particular, C.2A-29-31 and C.2A-35-36 reduced viability by approximately 99% (*p* < 0.0001) in both PBMCs and GIST-T1 cells. C.2A-48-49 and C.2A-60-61 showed a lower viability reduction compared to the previous ones and more prominent cytotoxicity in PBMCs than in GIST cells. Altogether, the observed cytotoxicity on PBMCs highlighted a non-specific mechanism of action. In this regard, doxorubicin, a chemotherapeutic drug successfully used to treat numerous solid tumors, is frequently associated with myelosuppressive side effects. In agreement with this, doxorubicin significantly impaired PBMC viability, similarly to FR2A clusters (Figure 9b), supporting the hypothesis that phytochemicals in the FR2A clusters could have chemotherapeutic properties.

### 2.6. Active Fractions Contain Pyrogallol-Bearing Compounds

The nine FR2-A active subfractions obtained from SEC were individually analyzed through ^1^H NMR to obtain an overview of the main phytochemical constituents. Additionally, UHPLC-MS was performed both in the support of NMR analysis and in an attempt to identify the molecules present at the lowest concentrations. The adjacent weakly or non-active subfractions were also phytochemically investigated. Interestingly, the analysis revealed the presence of pyrogallol-bearing compounds in all the active subfractions (Figure 10 and Appendix A). In particular, the active subfractions forming the cluster C.2A-29-31 harbor a pyrogallol derivative, whose structure remains to be not yet fully elucidated (Appendix A), together with a high quantity of fumaric acid; both were not found in the non-active subfractions. Regarding the subfractions forming the most potent cluster, C.2A-35-36, they predominantly contained gallic acid and the hexoside of the flavonoid myricetin, both bearing a pyrogallol moiety (Appendix A). Interestingly, the adjacent, weakly active subfraction—2A-33—also contained gallic acid, although at a lower concentration compared to the active cluster, supporting the involvement of this molecule in the weaker biological activity. The other adjacent subfraction—2A-39—mainly contained galloyl-arbutin, a pyrogallol derivative, and its weak activity could be attributed again to the pyrogallol moiety. Indeed, arbutin has been tested on GIST cells with negative results, and its non-anticancer activity was also confirmed by the absence of activity in the 2A-20 subfraction, which predominantly contained arbutin (Appendix A). Regarding the subfraction forming the cluster C.2A-48-49, the predominant molecule was gallocatechin, which also bears a pyrogallol moiety in its structure (Appendix A). The predominant phytoconstituent of the adjacent non-active subfraction 2A-47 was catechin (structurally similar to gallocatechin without the pyrogallol substituent). Hence, it could be hypothesized that the presence of the pyrogallol moiety is essential for the anticancer activity in GIST, a hypothesis supported once again by the analysis of the active cluster C.2A-60-61. Indeed, this cluster also presented a pyrogallol-bearing compound, trigalloyl-glucose, as the main phytoconstituent (Appendix A). In contrast, neither trigalloyl-glucose nor additional pyrogallol-bearing compounds were identifiable in the adjacent non-active 2A-58 subfraction.

In view of these results, gallic acid, bearing the pyrogallol moiety and already associated with anticancer activity [18], was tested in both GIST-T1 and GIST-882 cells, similarly to what was previously performed for β-arbutin; the 2A-35 subfraction was used as a positive control. Treatment with 30 µg/mL of gallic acid significantly impaired cell viability (*p* < 0.0001) similarly to the 2A-35 subfraction (Appendix A). Differently, the treatment with 6 µg/mL had no significant effect, while, as expected, 2A-35 significantly impaired the viability in both GIST cell lines (*p* < 0.0001) (Appendix A), suggesting that gallic acid is not sufficient, although it may promote an effect in concert with other phytochemicals.

## 3. Discussion and Conclusions

*Arbutus unedo* L. (Ericaceae family), known as the strawberry tree, is an evergreen shrub growing in circum-Mediterranean regions. Leaves are used in the traditional medicine of the Iberian Peninsula and Sardinia (Italy) to treat many illnesses [6], demonstrating a number of health-promoting properties associated with this plant. In this work, we showed that an extract from *Arbutus unedo* L. leaves (indicated as AUN) induces early apoptosis in GIST cells, exerting chemotherapeutic properties. This result is of great importance, considering chemotherapy is generally ineffective in advanced GISTs, and no options are available.

Patients’ treatment first relies on imatinib as the first-line therapy. Indeed, primary GISTs are commonly associated with gain-of-function mutations in KIT or PDGFRα (about 90% of cases), the disease drivers which, once mutated, show a ligand-independent dysregulated activity [19,20]. Moreover, given that secondary mutations in both KIT or PDGFRα have been found in imatinib-resistant GISTs, multi-target TKIs, such as sunitinib, regorafenib, and ripretinib, are administered as further treatment lines after the onset of resistance [12]. However, multi-target TKIs have not kept their promises, only modestly improving the outcomes of imatinib-resistant GIST patients. Indeed, imatinib-resistant subclones escape through alternative pathways that are not targetable from multi-target TKIs, suggesting that a TKI-based approach alone may not represent the only or best option for GIST patients [13]. Therefore, GIST patients urgently require the identification of novel therapies other than TKIs. In this context, we hypothesized that plant extracts could represent an important source of novel NPs to be studied in GIST. Among these, AUN showed antiproliferative activity in osteosarcoma (U2OS) cells [7], representing a promising unexplored therapeutic strategy in GIST. In view of this consideration, we tested the anticancer activity of AUN in GIST cellular models. AUN significantly impaired GIST cell viability, triggering a significant Annexin (+)/7-AAD (−) cell population, a marker of early apoptosis. AUN was further explored by bio-guided fractionation to deepen what could be the most active part of AUN. Hence, AUN was first fractionated by a liquid/liquid partition, separating roughly AUN phytochemicals on the basis of their polarity. This first step led to the identification of the FR2 fraction as the most active, able to impair cell viability and promote early apoptosis in different models of imatinib-sensitive GISTs (GIST-882 and GIST-T1), i.e., characterized by different primary kit mutations. In view of this consideration, we performed a second step of fractionation, subjecting the FR2 fraction to a reverse-phase MPLC, which led to the identification of the FR2-A fraction as the most active one in imatinib-sensitive cellular models (GIST-882 and GIST-T1), regardless of primary KIT mutations. Moreover, it resulted in being even more potent than FR2, as it significantly impaired cell viability at lower concentrations (12.5 μg/mL vs. 50 μg/mL). Afterward, FR2-A was tested in imatinib-resistant cell lines (GIST48 and GIST-48b), once again proving effective in impairing cell viability, providing new insight into the FR2-A mechanism of action. Indeed, the cell line GIST-48b is characterized by the absence of KIT expression, supporting the hypothesis that the mechanism is independent from KIT, possibly paving the way for developing therapeutic strategies other than TKIs in imatinib resistance. Additionally, FR2-A was more efficient with respect to imatinib in sensitive cells, inducing a higher percentage of Annexin V (+)/7-AAD (−) cells. This result confirmed the capacity of AUN to efficiently induce early apoptosis, prompting us to deepen the mechanism. Nevertheless, we did not observe the cleavage of caspase 3, caspase 9, and PARP-1, suggesting a cell death mechanism other than classical apoptosis. Interestingly, although not cleaved, a remarkable and rapid PARP-1 downregulation after FR2-A treatment was observed, possibly suggesting an epigenetic mechanism. Aware of the limitation, mainly due to the limited number of GIST cell models analyzed—though it depends on the availability of commercial cell lines—we are conscious that our results are speculative. However, recent findings in the literature seem to support our hypothesis of an epigenetic mechanism of action. Indeed, a study has identified PARP-1 as a target of miR-7-5p in a model of lung cancer cells, demonstrating that sponging miR-7-5p promotes the homologous repair (HR) path through upregulating PARP1 expression [21]. In another study, Garmutin-A (GA), an NP isolated from *Garcinia multiflora*, induced apoptosis via the upregulation of miR-17-5p, triggering PARP-1 downregulation, in leukemic CB3 cells [22]. Overall, this finding is in line with the evidence that many NPs could exert their anticancer activity by affecting miRNAs expression [23]. Therefore, a possible speculation is that FR2-A could upregulate different miRNAs, thereby altering the PARP-1-mediated repair capability. Another thought offered by PARP-1 downregulation is the possibility of re-thinking imatinib treatment in GIST, considering synergistic/combinatory approaches. Indeed, most GIST patients commonly achieve partial response or stable disease, suggesting that GIST cells could survive the treatment by activating adaptive responses. In this context, our study corroborates that imatinib mainly stabilizes GIST cells in a non-proliferative state rather than inducing cell death. Therefore, a combination of imatinib and PARP-1 inhibitors could represent a promising strategy that has not yet been investigated in GIST.

Regarding the possible bio-active compound characterizing FR2-A, we initially focused on β-arbutin. Indeed, this phytochemical, already identified as one of the primary metabolites in AUN [24], was identified as the most representative compound in FR2-A. Strengthened by this result, together with the knowledge that β-arbutin was already associated with proapoptotic activity in melanoma cells [25], we tested pure β-arbutin, unfortunately with negative results. This prompted us to perform an additional fractionation step, this time with a size-exclusion approach, to increase the concentrations of active phytochemicals. This approach led to sixty-one different subfractions, of which nine, forming four different clusters, exhibited a significant biological activity in GIST cells. Interestingly, all clusters contained phytochemicals characterized by the pyrogallol moiety, a polyphenolic structure already associated with proapoptotic activity. A study conducted on As4.1 juxtaglomerular cells, a model of benign kidney tumors, demonstrated that pyrogallol treatment could efficiently induce apoptosis [26]. Interestingly, the authors observed caspase 3 and PARP-1 cleavage; however, treatment with caspase 3 inhibitors did not prevent apoptosis, representing an intriguing result. Indeed, the authors did not evaluate the PARP-1 status in the presence of a caspase 3 inhibitor. Therefore, other mechanisms besides caspase activation could not be excluded. Another study demonstrated that the pyrogallol moiety in polyphenols had an important role to play in apoptosis induction in human embryonic kidney cells (HEK293T) and the chronic myelogenous leukemia cell line (K562), even though the mechanism was not investigated [27]. In the same study, the authors highlighted that the pyrogallol moiety was important for cytotoxic activity. In agreement with this, all the FR2-A active subfractions showed cytotoxicity in PBMCs, similar to those observed with doxorubicin, a traditional chemotherapeutic. These findings suggest a non-targeted but chemotherapy-like mechanism of action, which should not be underestimated considering the lack of effective chemotherapeutic agents for GIST patients. Indeed, doxorubicin, approved in several solid tumor treatments, has been tested in clinical trials with poor response in GISTs, results that led to its non-approval [14]. In confirmation of this, unlike FR2-A clusters, doxorubicin does not impair the viability of imatinib-resistant cells [28]. Therefore, deepening FR2-A composition could be of great interest in GIST, leading to the identification of novel chemotherapeutic agents to be used as alternatives or in combination with the approved TKI. In view of these considerations, the most active FR2-A subfractions (2A-35 and 2A-36) were mainly characterized by the presence of gallic acid, a phenolic acid which has been identified as a promising health-promoting agent in many conditions, including cancer [29]. Despite the expectations, our finding ascribed an activity to gallic acid, which, however, was weaker than 2A-35. Therefore, gallic acid cannot be considered a solely bioactive compound but rather one that acts in concert with other components, highlighting the importance of the phytocomplex.

In conclusion, although bio-guided fractionation did not lead to the identification of a single compound active in GIST cellular models, it was confirmed as a feasible tool for circumscribing phytochemicals of interest. In particular, in the present study, we identified pyrogallol-bearing compounds as the main players in GIST cell cytotoxicity, harboring chemotherapeutic properties in imatinib-sensitive and resistant cells. These data may fuel further studies searching for novel treatments in GIST, including additional strategies that augment the existing treatments, resulting in better patient outcomes. Therefore, the chemotherapeutic-like activity should not discourage further preclinical studies on pyrogallol-bearing compounds in GISTs to conclude whether they could represent an alternative strategy for treating GISTs.

## 4. Material and Methods

### 4.1. GIST Cell Lines and Cell Culture Conditions

GIST-882 and GIST-T1 are defined as primary mutated and imatinib-sensitive cellular models. They harbor exon 9 (K642E) homozygous mutations and exon 13 heterozygous (V560-Y579) in the KIT gene, respectively. GIST-882 and GIST-T1 were grown in RPMI-1640 supplemented with 15% FBS. GIST-48 is instead reported as an imatinib- and sunitinib-resistant cell line harboring a primary homozygous mutation on KIT exon 11 (V560D) and an additional secondary heterozygous mutation in exon 17 (D820A). GIST-48b was established in vitro, starting from GIST-48 after HSP-90 inhibitor (17-AAG) drug pressure selection, resulting in a subline characterized by nearly undetectable KIT transcript and protein. GIST-48 and GIST-48b were grown in Iscove’s Modified Dulbecco’s Medium (IMDM) supplemented with 15% FBS. All the indicated cell lines were routinely tested to avoid mycoplasma contamination with MycoBlue Mycoplasma Detector (Vazyme, Nanjing, China). GIST-T1 was purchased from CosmoBio (Tokyo, Japan), while GIST-48 and GIST-882 cells were kindly provided by Fletcher JA, MD (Harvard Medical School).

### 4.2. Plant Material

The leaves of *Arbutus unedo* L. were harvested in Sardinia in 2018 (voucher specimen CAG 878/v, deposited at the General Herbarium of the Department of Life and Environmental Sciences, University of Cagliari). To ensure reproducibility, the plant material was analyzed by ^1^H NMR profiling [25].

### 4.3. Cell Viability and Apoptotic Profile

10^5^ GIST cells were seeded in a 24-well cell culture plate the day before treatment. Cells were treated with AUN or AUN subfractions at the indicated final concentrations in the culture medium for the indicated time points. The analysis of cell viability and the presence of apoptotic cells were evaluated on the entire cellular population, including cells that were in suspension due to the treatment. For this reason, the culture medium and the phosphate-buffered saline (PBS) used to rinse cells before trypsinization were also collected. Adherent cells were harvested by trypsinization and combined with cells in the culture medium and PBS. The entire sample was then centrifugated and resuspended in 500 µL of fresh medium. Samples were stained with Guava^®^ ViaCount™ or Guava^®^ Nexin Reagent according to the manufacturer’s instructions. The Guava^®^ ViaCount™ distinguishes viable and non-viable cells based on the differential permeabilities of two DNA-binding dyes. The nuclear dye (in the graphs indicated as “Nucleated Cells” on the y-axis) differentiates nucleated cells from cellular debris, while the viability dye brightly stains dying cells (in the graphs indicated as “Viability” on the x-axis). The number of viable cells in the sample was defined starting from the cell concentration (cells/μL) calculated by the flow cytometer instrument. The monitoring of apoptosis activation was performed using Guava^®^ Nexin Reagent. The reagent combines fluorescently labeled Annexin V (Annexin V-PE) and 7-Aminoactinomycin D (7-AAD). Annexin V-PE, a calcium-dependent phospholipid-binding protein with a high affinity for phosphatidylserine (PS), is a membrane component early exposed on the external cell surface during apoptotic pathway stimulation (indicated in the graphs as “Annexin-V” on the x-axis). Instead, late apoptotic/dead cells are recognized by 7-AAD. This fluorescent DNA intercalator is membrane-impermeant and excluded from live, healthy, and early apoptotic cells (indicated in the graphs as “Viability” on the y-axis).

### 4.4. AUN Extraction and Fractionation Steps of A. unedo

Five hundred grams of dried and grounded plant material was extracted using 2 L of CH_3_OH 80% *v*/*v*. The extract was filtered and dried in a rotary evaporator; the extraction procedure was repeated four times (24 h each) on the same plant material, obtaining 67.7 g of extract. The extract (AUN) was suspended in 700 mL of water and subjected to liquid/liquid partition using in-series chloroform (CHCl_3_) or ethyl acetate (EtOAc). Phytochemical extractions were performed using each solvent three times. Three AUN-derived fractions were obtained: CHCl_3_ (FR1; yield: 0.33% *w*/*w*), EtOAc (FR2; yield: 8.8% *w*/*w*), and H_2_O (FR3; yield 70%). The stock solutions for the bioassays of FR3 were prepared by solubilizing dried extracts in distilled H_2_O at an initial concentration of 20 mg/mL, while FR2 and FR3 were instead solubilized in DMSO 10% at the same initial concentration. FR1, FR2, and FR3 fractions were then diluted (1:1000) during the biological assays, leading both H_2_O and DMSO to a final concentration of 0.1% in the medium supplemented with FBS 15%. Being the bioactivity localized in the FR2, 4 g of the fraction was suspended in 5 mL of water and injected in a medium-pressure liquid chromatography (MPLC) instrument (Reveleris^®^, Büchi, Switzerland) using a reverse-phase stationary phase (40 g of C_18_ column). A gradient of water (solvent A) and methanol (solvent B) was used as an eluent. The gradient was composed of an isocratic phase of 10 min (90% A and 10% B), a gradient from 90% A to 80% A in 1.1 min, an isocratic phase of 20 min (80% A and 20% B), a gradient from 80% A to 70% A in 1.1 min, an isocratic phase of 10 min (70% A and 30% B), a gradient from 70% A to 50% A in 1.1 min, an isocratic phase of 10 min (50% A and 50% B) a gradient from 50% A to 30% A in 1.1 min, an isocratic phase of 5 min (30% A and 70% B), a gradient from 30% A to 0% A in 1.1 min, an isocratic phase of 5 min (0% A and 100% B). The flow rate was 20 mL/min, and the run length was 70 min. Based on the chromatogram and UV–Vis absorbance signals at three wavelengths (UV1 = 254 nm, UV2 = 270 nm, and UV3 = 340 nm), FR2-derived fractions were collected and dried, obtaining a total of four FR2 subfractions (FR2-A, FR2-B, FR2-C, and FR2-D). An analogous preparation of FR2 was followed for its subfractions in the biological assays. Being the bioactivity localized in the FR2-A, 879.8 mg of the fraction was suspended in the minimum amount of methanol and then subjected to size exclusion chromatography using a chromatography column (1800 mm × 25 mm) filled with 220 g of Sephadex (LH-20) and, as an eluent, methanol. The flow rate was 0.4 mL/min. The eluate in each tube was concentrated in a rotary evaporator, and a small quantity was used to perform thin-layer chromatography (TLC). The stationary phase of the TLC used a silica gel matrix with a 254 nm florescent indicator (Sigma-Aldrich), while EtOH:MeOH:H_2_O (10:1.35:1) was used as the mobile phase. This latter was employed to acquire a first overview of the chemical composition of the fractions by UV–Vis light exposure, enabling us to group the most similar ones, obtaining 84 final fractions. Excluding the fraction yielding less than 1 mg of material, 61 were tested in the bioassays. The most active ones were phytochemically investigated compared to the inactive or weakly active ones, which were eluted immediately before or after.

### 4.5. NMR Spectra Measurement

^1^H NMR spectra were recorded at 25 °C on a Varian Inova instrument (equipped with a reverse triple resonance probe) operating at 600.13 MHz. Each ^1^H NMR spectrum consisted of 256 scans (corresponding to 16 min) with a relaxation delay (RD) of 2 s, acquisition time of 0.707 s, and spectral width of 9595.8 Hz (corresponding to δ 16.0). All the ^1^H NMR spectra were uploaded onto Zenodo.

### 4.6. UHPLC-MS Analysis

The UHPLC-MS analysis was run on a Waters ACQUITY ARC UHPLC/MS system consisting of a QDa mass spectrometer equipped with an electrospray ionization interface and a 2489 UV–Vis detector. The detected wavelengths (λ) were 254 nm and 365 nm. The analyses were performed on an XBridge BEH C18 column (10 × 2.1 mm i.d., particle size 2.5 µm) with an XBridge BEH C18 VanGuard Cartridge precolumn (5 mm × 2.1 mm i.d., particle size 1.8 µm). The mobile phases were H_2_O (0.1% formic acid) (A) and MeCN (0.1% formic acid) (B). Electrospray ionization in positive and negative modes was applied in the 50–1200 Da mass scan range. The FR2-A subfractions were diluted to 100 µg/mL, and a volume of 3 µL was injected. The samples were eluted with the following method: 20% B for one minute, followed by a gradient reaching 95% B in 3 min; 95% B was kept for 1 min; then the gradient reached 20% B in 0.2 min again; and 20% B was kept for 2 min. The flow rate was 0.8 mL/min.

### 4.7. Software and Statistical Analysis

Statistical analysis was performed using GraphPad Prism software 8.0.2, applying the appropriate statistical test. Details are indicated below each figure. IC50 was instead calculated with the free “aatbioquest” tool (https://www.aatbio.com/tools/ic50-calculator, accessed on 11 April 2024).

### 4.8. Western Blot

Whole-cell protein lysates were prepared using NP40 buffer containing protease inhibitors (halt protease and phosphatase inhibitor cocktail; Thermo Fisher Scientific) and 1 mM of phenylmethylsulfonyl fluoride (Sigma-Aldrich, St. Louis, MI, USA). Proteins were separated in SDS-PAGE (12%) and transferred onto nitrocellulose membranes. Membranes were blocked by 5% skimmed milk, followed by incubation at 4 °C overnight with primary antibodies. Primary antibodies against KIT (A4502; Dako, Glostrup, Denmark), phospo-KIT (3391; Cell Signaling, Danvers, MA, USA), PARP-1 (9542, Cell-Signaling), and actin (A1978; Sigma-Aldrich) were used. After rinsing, membranes were incubated with horseradish peroxidase-conjugated secondary antibody (Thermo Fisher Scientific) at room temperature for 2 h. After further rinsing, immunoreactive bands were visualized by enhanced chemiluminescence (BioRad, Hercules, CA, USA), and signals were captured and quantified using ChemiDoc (BioRad).

### 4.9. Chemical Compounds

Sigma-Aldrich supplied β-arbutin and gallic acid for the biological assays. Selleck Chemicals supplied imatinib.

### 4.10. Cell Viability by MTT

2.5 × 10^4^ cells were seeded in a 96-well culture plate the day before the treatment. Cells were treated with FR2-A-derived fractions, and viability was analyzed 24 h after treatment. Treatment was removed, and cells were incubated with MTT reagent (0.5 mg/mL) in a medium without serum for 2 h. At the end of the incubation, the MTT solution was removed carefully, and formazan crystals were dissolved in DMSO. Absorbance was read at 492 nm using a TECAN spectrophotometer.

### 4.11. PBMC Isolation

Peripheral blood lymphocyte (PBMC) isolation was performed as described [30]. Briefly, PBMCs were isolated from blood samples of healthy donors (Buffy coat) using a density gradient centrifugation with Histopaque-1077. Donors were healthy, non-smokers, and with no known exposure to genotoxic chemicals or radiation. Authorization for the use of human blood samples for research purposes was received from AUSL Bologna IT, S. Orsola-Malpighi Hospital.

## Figures and Tables

**Figure 1 plants-13-01201-f001:**
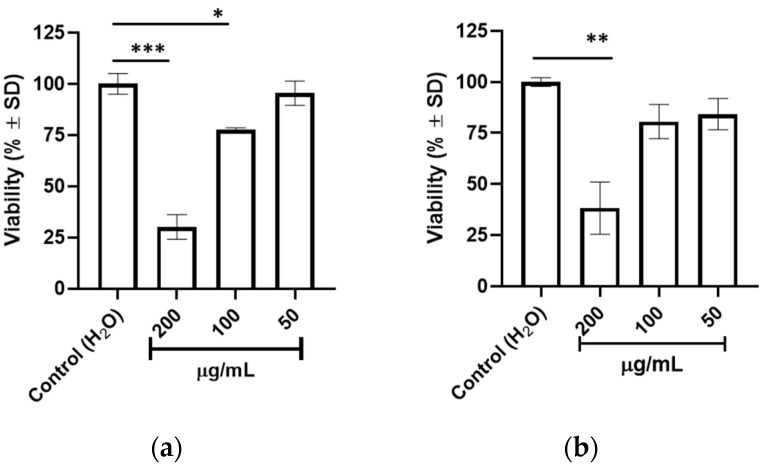
Effect of AUN on GIST cell viability. GIST-882 (**a**) and GIST-T1 (**b**) cells were treated with different concentrations (50, 100, 200 µg/mL) for 72 h. Viability is expressed as percentage (%) ± standard deviation (SD). Adjusted *p*-value * <0.05, ** <0.01, *** <0.001 (one-way ANOVA–Dunnett’s multiple comparison test with respect to control). Number of independent experiments (*n*) = 3.

**Figure 2 plants-13-01201-f002:**
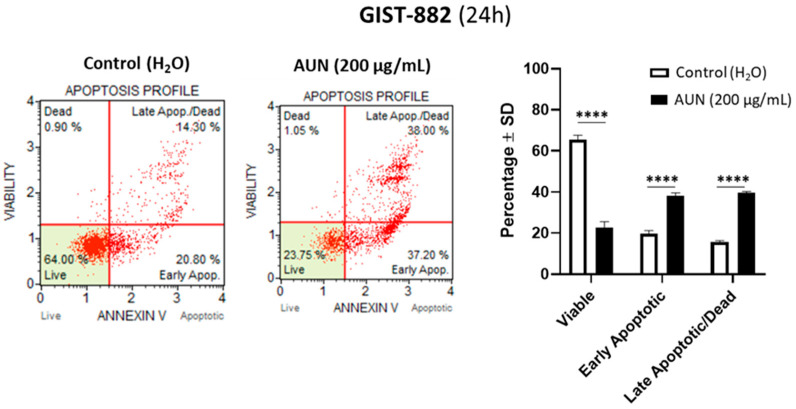
AUN promotes the rise of early and late apoptotic cell populations in GIST cells. Apoptotic profile of AUN-treated GIST-882 (24 h) (**upper panel**) and GIST-T1 (6 h) (**lower panel**) cells with respect to control (representative examples). Apoptotic profiles are defined by Annexin-V/7-AAD assay. The % ± SD of viable (Annexin-V (−)/7-AAD (−)), early apoptotic (Annexin V (+)/7-AAD (−)), and late apoptotic/dead (Annexin V (+) and 7-AAD (+)) cell populations is displayed. Adjusted *p*-value *** <0.001, **** <0.0001 (multiple *t*-test). N = 3.

**Figure 3 plants-13-01201-f003:**
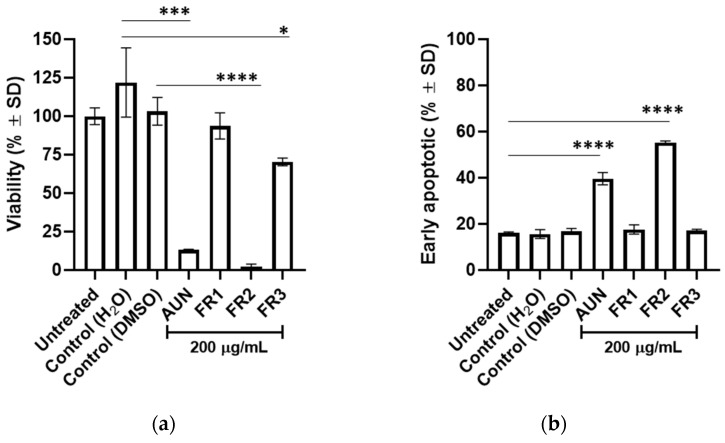
Effect of AUN and its fractions on GIST-882 cells. (**a**) Cell viability after 24 h. Viability is expressed in % ± SD. Adjusted *p*-value * <0.05, *** <0.001, **** <0.0001 (one-way ANOVA–Tukey’s multiple comparison with respect to control). (**b**) % ± SD of early apoptotic (Annexin V (+)/7-AAD (−)) cells is displayed. Adjusted *p*-value **** <0.0001 (one-way ANOVA–Dunnett’s multiple comparison with respect to control). N = 3.

**Figure 4 plants-13-01201-f004:**
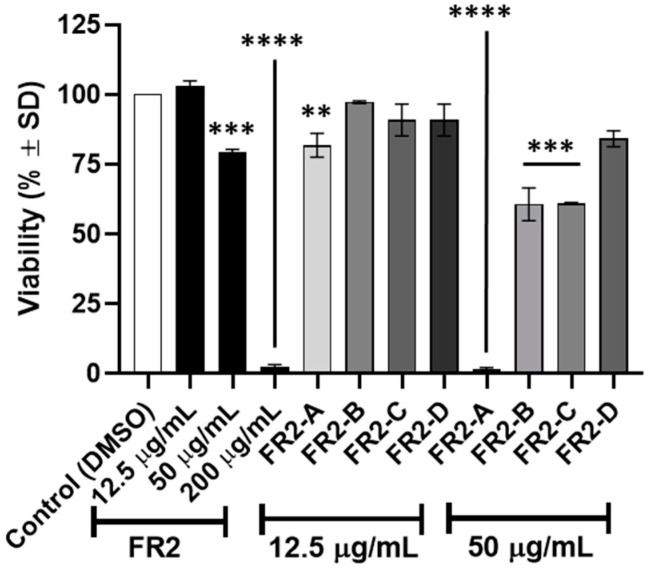
Effect of FR2 and its subfractions on cell viability. GIST-882 cells were treated with FR2 or its derived subfraction for 24 h at the indicated final concentration (µg/mL). Viability is expressed in % ± SD. Adjusted *p*-value ** <0.01, *** <0.001, **** <0.001, (one-way ANOVA–Tukey’s multiple comparison test with respect to control). N = 3.

**Figure 5 plants-13-01201-f005:**
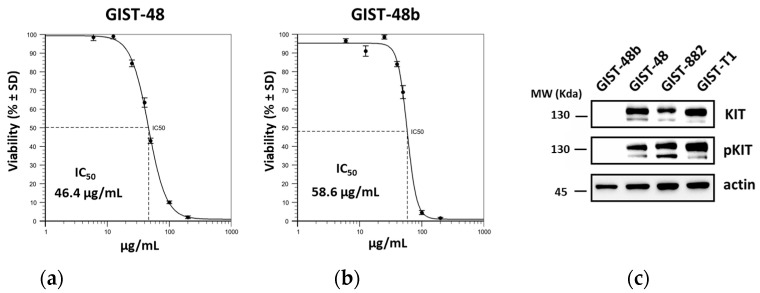
FR2-A IC50 in imatinib-resistant GIST-48 and GIST-48b. GIST-48 (**a**) and GIST-48b (**b**) were treated with different concentrations of FR2-A. Cell viability was measured through the Guava^®^ ViaCount™ staining and flow cytometry approach. Cell viability is expressed in % ± SD. The calculated IC50 is reported at the bottom of the corresponding graph. A representative experiment among two replicates is shown. (**c**) Expression of total and activated KIT in GIST cell lines by Western blot. The expression level of actin was used as the reference. The molecular weight (MW) is reported.

**Figure 6 plants-13-01201-f006:**
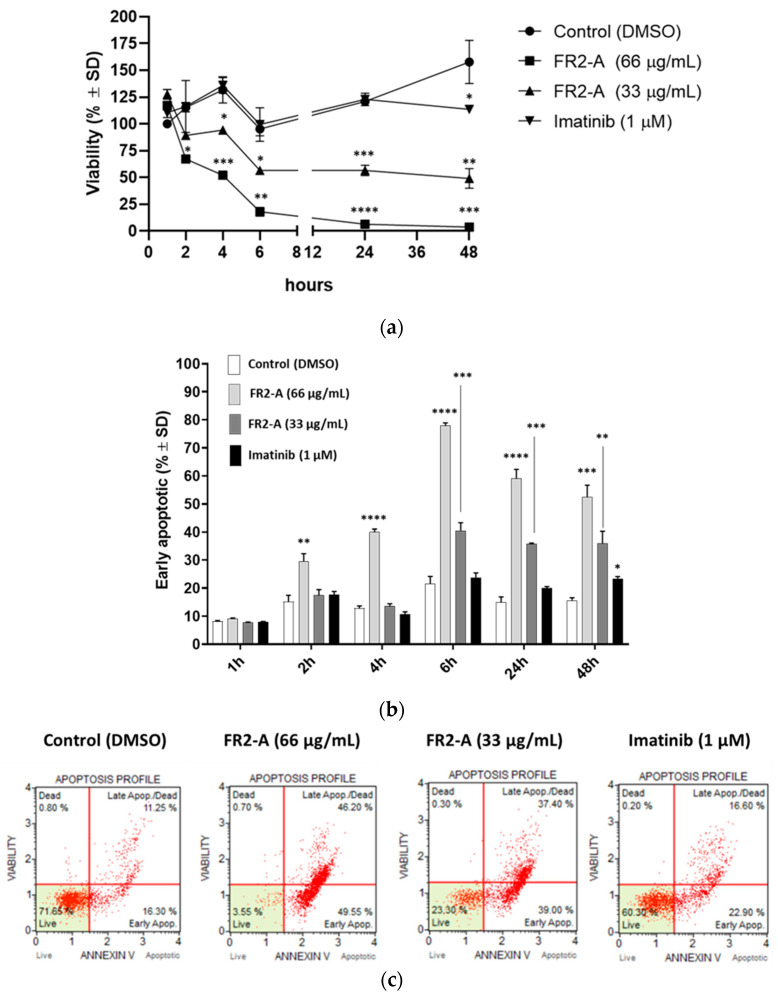
FR2-A and imatinib-treated samples in a 48 h time-course experiment. (**a**). Cell viability (% ± SD). Adjusted *p*-value compared to control: * <0.05, ** <0.01, *** <0.001, and **** <0.0001 (one-way ANOVA and Dunnett’s multiple comparison test). (**b**) Percentage ± SD of early apoptotic (Annexin V (+)/7-AAD (−) cell populations. Adjusted *p*-value compared to control: * <0.05, ** <0.01, *** <0.001, and **** <0.0001 (one-way ANOVA and Dunnett’s multiple comparison). (**c**) Flow cytometry analysis (raw data) of Annexin V and 7-AAD staining in control, 66 µg/mL FR2-A, 33 µg/mL FR2-A, and 1 µM imatinib-treated samples at 48 h. N = 3.

**Figure 7 plants-13-01201-f007:**
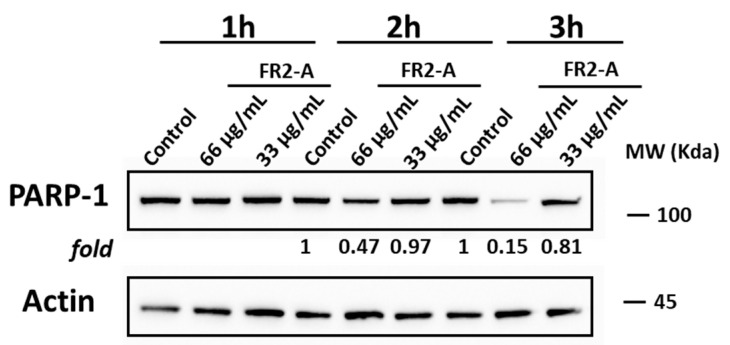
PARP-1 expression following FR2-A treatment. The expression level of PARP-1 after FR2-A (IC50 and 2-times IC50 doses) treatment at 1 h, 2 h, and 3 h is shown. Actin was used as a control. Data are expressed as the fold based on the actin-related level with respect to the corresponding control samples at each time point. A representative experiment among two replicates is shown.

**Figure 8 plants-13-01201-f008:**
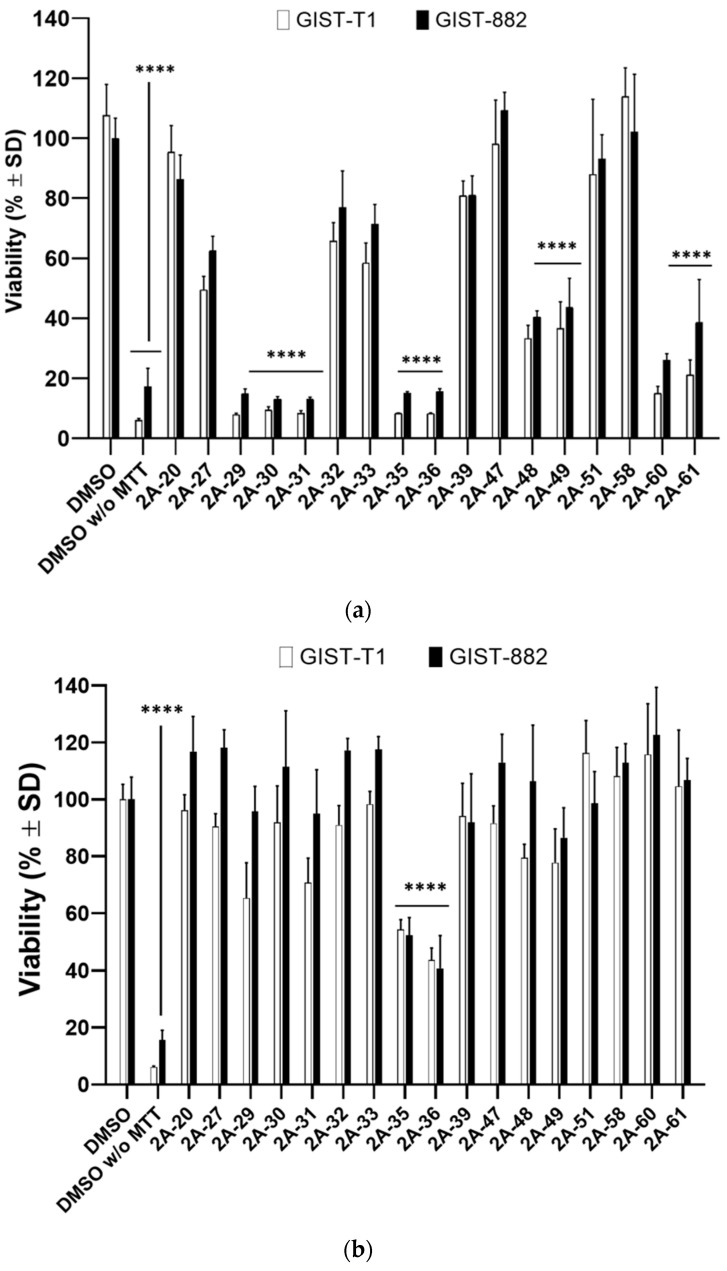
FR2-A subfractions significantly impaired GIST-882 and GIST-T1 viability. Cell viability was assessed 24 h after treatment with 30 µg/mL (**a**) and 6 µg/mL (**b**) using the MTT assay. The viability is expressed in % ± SD; adjusted *p*-value compared to DMSO: **** <0.0001 (one-way ANOVA and Dunnett’s multiple comparison test). DMSO w/o MTT means DMSO control sample without the addition of MTT reagent, thus representing the assay limit of detection. N = 3.

**Figure 9 plants-13-01201-f009:**
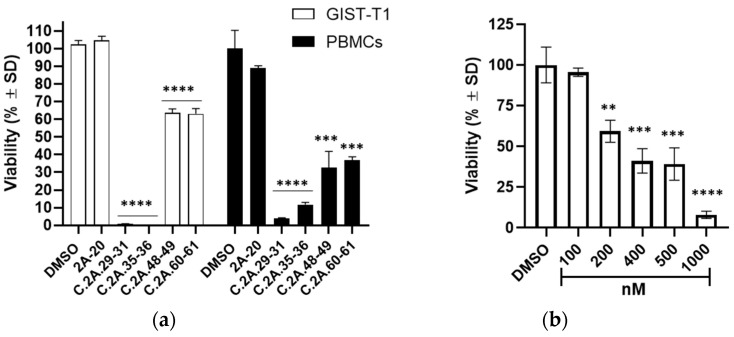
In vitro toxicity of FR2-A clusters (a) GIST-T1 and PBMCs cell viabilities after treatment with 2-A selected clusters (30 µg/mL). 2A-20 subfraction was used as a negative control (30 µg/mL). Viability is expressed in % ± SD. (**b**) PBMC viability after treatment with different doses of doxorubicin. Viability is expressed in % ± SD. For both adjusted *p*-value compared to DMSO: ** <0.01, *** <0.001, and **** <0.0001 (one-way ANOVA and Dunnett’s multiple comparison test). N = 3.

**Figure 10 plants-13-01201-f010:**
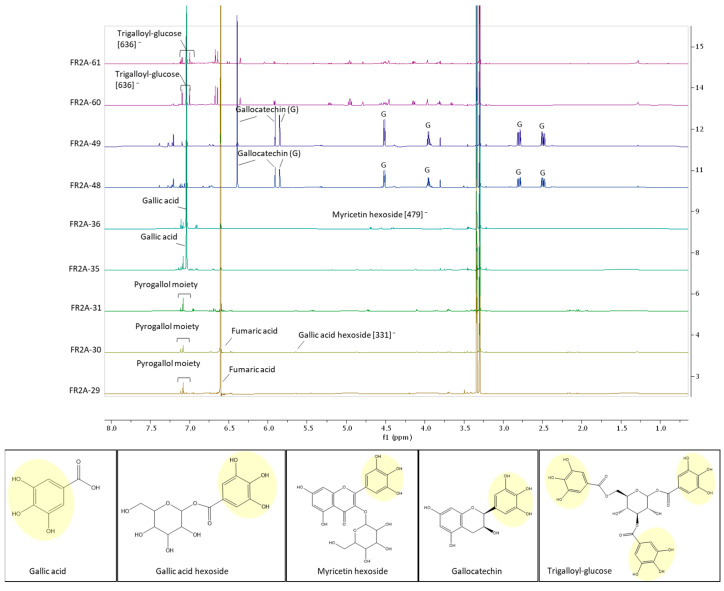
Phytochemical analysis of the bioactive FR2-A subfractions. The ^1^H NMR spectrum in MeOH-d4 of each subfraction is reported. In addition, the m/z value of the compounds emerging by ESI-MS analysis (in negative mode) is reported in square brackets. The molecular structure of each putative active compound is represented with the commonly found pyrogallol moiety highlighted in yellow.

## Data Availability

The spectral data are available in the Zenodo repository [DOI:10.5281/zenodo.10868280]. The bioactivity data are available on request from the corresponding author.

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
