# Peer review of "Arbutus unedo* L. Fractions Exhibit Chemotherapeutic Properties for the Treatment of Gastrointestinal Stromal Tumors"

_plants, 2024, doi:10.3390/plants13091201_

Round 1
Reviewer 1 Report
Comments and Suggestions for Authors
The subject of the study is very interesting, the work is well designed and the results and conclusions are meaningful. The paper, which is very well written, is worth of publication.
I only suggest some minor corrections.
- “A. unedo” must always be in italics.
- IC50: the “50” should be in subscript.
- Line 68: AUN is the acronym used by the authors for Arbutus unedo L.. When referring to the hydroalcoholic crude extract this should be specified for clarity (e.g. “AUN’s hydroalcoholic crude extract impairs GIST cell viability and triggers apoptosis”): Alternatively, use a different acronym for this crude extract.
- The operating frequency of the NMR spectrometer should be provided.
Author Response
Dear Reviewer 1,
We are honored that you have considered our study interesting, well-designed, and meaningful. We thank you for the time you have dedicated to peer review. Our responses are below in red.
- “A. unedo” must always be in italics
We have corrected it as you have indicated.
- IC50: the “50” should be in subscript.
We thank you for your suggestion, but we have commonly found it in literature as IC50 (Caldwell, Gary W et al. 2012*). Therefore, if it is not a problem with editorial policy, we would like to maintain this form.
*Caldwell, Gary W et al. “The IC(50) concept revisited.” Current topics in medicinal chemistry vol. 12,11 (2012): 1282-90. doi:10.2174/156802612800672844
- Line 68: AUN is the acronym used by the authors for Arbutus unedo L.. When referring to the hydroalcoholic crude extract this should be specified for clarity (e.g. “AUN’s hydroalcoholic crude extract impairs GIST cell viability and triggers apoptosis”): Alternatively, use a different acronym for this crude extract.
We used AUN as the acronym for the “hydroalcoholic crude extract” obtained from the leaves. However, as you suggested, we used the acronym in the title of the first paragraph (Line 68) without explaining this before. In fact, we only described it in Lines 69-70. For this reason, according to your suggestion, we have changed the title of the first paragraph while maintaining the use of AUN as the acronym of the hydroalcoholic crude extract.
- The operating frequency of the NMR spectrometer should be provided.
The operating frequency of the NMR spectrometer is provided in the materials and methods (Line 522). We also added the reference to the Zenodo repository (Line 587).
Reviewer 2 Report
Comments and Suggestions for Authors
Abstract
The aim of the study should be indicated. The conclusion in this section should be rewritten. In vivo and other toxicological studies are necessary to conclude whether these fractions can be a treatment for gastrointestinal stromal tumors.
Introduction
The pharmacological and toxicological studies with Arbutus unedo should be mentioned. In addition, the ethnomedicinal uses of Arbutus unedo should be incorporated in this section.
Materials and methods
A positive control (reference drug) of cytotoxicity and apoptosis should be included.
Results
Figures 1, 4, 5, 8, and 9 should be erased. The results can be presented as IC50 values.
Discussion
Add a brief description of KIT in the pathogenesis of gastrointestinal cancer.
References
The format of references is not homogenous. Check and correct the format of references (e.g., 8, 11, 23, and others), when necessary.
Comments on the Quality of English Languageno comments
Author Response
Dear Reviewer 2,
We sincerely thank you for your dedicated time reviewing our research article. We particularly appreciate your points of view, which allow us to improve all sections of the manuscript. Our responses are below in red.
Abstract
The aim of the study should be indicated. The conclusion in this section should be rewritten. In vivo and other toxicological studies are necessary to conclude whether these fractions can be a treatment for gastrointestinal stromal tumors.
A new abstract version has been written to explain the study's aim better and provide conclusions more clearly. We also added the sentence you suggested to the abstract and in the conclusion of our manuscript (Lines 436-438).
Introduction
The pharmacological and toxicological studies with Arbutus unedo should be mentioned. In addition, the ethnomedicinal uses of Arbutus unedo should be incorporated in this section.
We have mentioned the ethnomedicinal uses of Arbutus unedo L. and studies investigating the pharmacological/toxicological effect of its derived extracts (Lines 38-47). The introduction has been lightly modified to include and properly integrate the new information with the text.
Materials and methods
A positive control (reference drug) of cytotoxicity and apoptosis should be included.
A positive control (reference drug) of cytotoxicity and apoptosis in GIST cannot be provided since imatinib and TKIs, the approved therapies, mostly promote the stabilization of the disease, preventing cell proliferation rather than inducing intensive apoptosis (Gupta et al. 2010*). However, as shown in our comparison between imatinib and FR2-A in GIST-882 (Figure 6C), imatinib promotes a faint increase in the population, which is positive for the Annexin V signal, thus also representing a positive control of the rise of these apoptotic populations. The Annexin V assay is historically well-recognized for identifying early/late apoptotic cells (van Engeland, M et al. 1998#).
* Gupta, Anu et al. “Autophagy inhibition and antimalarials promote cell death in gastrointestinal stromal tumor (GIST).” Proceedings of the National Academy of Sciences of the United States of America vol. 107,32 (2010): 14333-8. doi:10.1073/pnas.1000248107
#van Engeland, M et al. “Annexin V-affinity assay: a review on an apoptosis detection system based on phosphatidylserine exposure.” Cytometry vol. 31,1 (1998): 1-9. doi:10.1002/(sici)1097-0320(19980101)31:1<1::aid-cyto1>3.0.co;2-r
Results
Figures 1, 4, 5, 8, and 9 should be erased. The results can be presented as IC50 values.
We are sorry, but what you suggest with “Figures 1, 4, 5, 8, and 9 should be erased” is unclear. We have modified IC50 in Figure 5, which was wrongly reported with “50” in the subscript.
Discussion
Add a brief description of KIT in the pathogenesis of gastrointestinal cancer.
A brief description of KIT and PDGFRα has been included in the discussion (Line 339-344).
References
The format of references is not homogenous. Check and correct the format of references (e.g., 8, 11, 23, and others), when necessary.
The format of the references was made with the tool provided by “Mendeley Cite”. In particular, we selected a format that allows us to cite publications with a number-based list, as indicated in the author guidelines. Certain citations are indicated with the format “et al.” because of numerous authors. We checked, and the articles we mentioned are appropriately cited.
Reviewer 3 Report
Comments and Suggestions for Authors
The paper “Arbutus unedo L. fractions exhibit chemotherapeutic properties
for the treatment of gastrointestinal stromal tumors”
conducted by Aldo Di Vito et al. provides information on the effect of natural extract in antitumor treatment (GIST).
The study highlighted the fact that through the bioguided fractionation of a hydroalcoholic extract obtained from the leaves of Arbutus unedo L., a subfraction (FR2-A) was obtained, which proved action on the viability of cells sensitive or resistant to imatinib.
The subject is relevant for research, because it brings valuable information about possibility of treatment in GIST. What is relevant, is the rapid reduction effect of PARP-1, without the traditional caspase-dependent cleavage.
The data presented show that FR2-A targets GIST cells through a non-specific mechanism of action probably promoted by pyrogallol-carrying compounds, as part of a phytocomplex.
The conclusion of the study was that the obtained FRs showed chemotherapeutic properties in GIST cells, encouraging further studies of Arbutus unedo L. Fractions derived from pyrogallol-bearing compounds can be considered an alternative way to treat GIST.
The formulated conclusions are in accordance with the obtained results, based on the proposed study objectives.
The references are recent and relevant to the information presented.
Author Response
Dear Reviewer 3,
We are thankful for your review. We are glad that you found the presented results relevant to GIST research and that you also considered our conclusions in accordance with the results.
Round 2
Reviewer 2 Report
Comments and Suggestions for Authors
Doxorubicin, docetaxel, 5-fluorouracil, and others, are drugs used for gastrointestinal cancer, and at least one of these can be used as a reference drug for the study. The results of cytotoxicity and apoptosis of this study should compared with a positive control.
Figures 1, 4, 5, 8, and 9 should not be presented in the manuscript. The results shown in these figures can be presented only as IC50 values. These figures are unnecessary.
Comments on the Quality of English Languageno comments
Author Response
Dear Reviewer 2,
We thank you for the time you dedicated to our manuscript and for the comments. We retain that your point of view could be due to our inadequate description/explanation of chemotherapeutics in GISTs, a subset of gastrointestinal malignancies. We mentioned these elements in our introduction but probably not in a sufficient and well-described way. We have integrated the manuscript with more information (Introduction Lines 57-74). We hope that in this form, you agree with us regarding the impossibility of using reference drugs/positive controls you mentioned in our study.
You can find a detailed response below.
Doxorubicin, docetaxel, 5-fluorouracil, and others, are drugs used for gastrointestinal cancer, and at least one of these can be used as a reference drug for the study. The results of cytotoxicity and apoptosis of this study should compared with a positive control.
The manuscript focuses only on Gastrointestinal stromal tumors (GISTs), a subset of gastrointestinal malignancies. GISTs are not responsive to chemotherapeutic agents, including those mentioned by the reviewer (Doxorubicin, docetaxel, 5-fluorouracil, etc.), and they are not approved for the clinical management of GISTs. Indeed, the therapy of GISTs only relies on TKI-based targeted therapy, with imatinib as the first-line treatment. De Matteo et al. (2002)1 in Table 2 reported that doxorubicin and docetaxel did not improve partial response in patients with an unresectable or metastatic GIST. In agreement with these clinical data, in a broad drug repurposing preclinical study, doxorubicin, docetaxel, 5-fluorouracil, and further 793 U.S. Food and Drug Administration (FDA)-approved drugs, including further traditional chemotherapeutics, were not found to be active in both imatinib-sensitive and -resistant GIST cells (Pessetto et al. 20132). Therefore, in this context, none of these drugs could represent a reliable reference drug/positive control to induce cytotoxicity and apoptosis in our study. According to the literature, these compounds could instead represent negative controls since they don’t efficiently target imatinib-sensitive and resistant GIST cells and are not used in clinics.
We used imatinib as a reference drug instead since it is the first-line therapy, demonstrating that FR2-A can quickly and more efficiently reduce cell viability, promoting the rise of a bigger Annexin V-positive population (Figure 6). As expected by preclinical findings, imatinib can mostly prevent the proliferation of GIST cells rather than stimulate intense apoptosis. However, the apoptosis profile shown in Figure 6 allows us to observe a small population of Annexin-positive cells, a positive control of apoptosis. Regarding cytotoxicity in healthy cellular models, we used doxorubicin as a positive control in PBMCs since its myelosuppressive side effects are well-recognized in clinics.
- Dematteo, Ronald P et al. “Clinical management of gastrointestinal stromal tumors: before and after STI-571.” Human pathology 33,5 (2002): 466-77. doi:10.1053/hupa.2002.124122
- Pessetto, Ziyan Y et al. “Drug repurposing for gastrointestinal stromal tumor.” Molecular cancer therapeutics 12,7 (2013): 1299-309. doi:10.1158/1535-7163.MCT-12-0968
Figures 1, 4, 5, 8, and 9 should not be presented in the manuscript. The results shown in these figures can be presented only as IC50 values. These figures are unnecessary.
We only treated cells with few experimental concentrations of AUN, FR2-B, FR2-C, FR2-D, and FR2-A subfractions and established clusters, respectively (Figures 1,4,5,8 and 9). Therefore, the used concentrations are insufficient to calculate a reliable IC50. We only calculated the IC50 of FR2-A in imatinib-sensitive (GIST-882 and GIST-T1 – Supplementary Figure 4) and in imatinib-resistant cells (GIST-48 and GIST-48b – Figure 5). Although, as you suggested, showing the dose-response curve could be unnecessary, if it is not a problem for the Editor and/or is not a problem for journal policy, we would like to present it. We think that in this way, the reliability and understanding of the results could be more accessible for any readers. More generally, our aim was not to calculate the IC50 of AUN and derived subfractions but to identify novel compounds that could be further studied in GISTs. We calculated the IC50 of FR2-A only to assess its high-broad spectrum activity in both imatinib-sensitive and -resistant cells.
Sincerely,
Professor Sabrina Angelini
Round 3
Reviewer 2 Report
Comments and Suggestions for Authors
The manuscript can be accepted for publication. However, the authors should mention in the manuscript the reason for not using Doxorubicin, docetaxel, 5-fluorouracil or other drugs in this work.
Comments on the Quality of English Languageno comments